# Percutaneous Thermal Ablation for Renal Tumors in Patients with Birt–Hogg–Dubé Syndrome

**DOI:** 10.3390/cancers14204969

**Published:** 2022-10-11

**Authors:** Sylvain Bodard, Idris Boudhabhay, Charles Dariane, Christophe Delavaud, Sylvain Guinebert, Dominique Joly, Marc-Olivier Timsit, Arnaud Mejean, Virginie Verkarre, Olivier Hélénon, Stéphane Richard, Jean-Michel Correas

**Affiliations:** 1AP-HP, Hôpital Necker Enfants Malades, Service d’Imagerie Adulte, F-75015 Paris, France; 2Université de Paris Cité, F-75006 Paris, France; 3Sorbonne Université, CNRS, INSERM Laboratoire d’Imagerie Biomédicale, F-75005 Paris, France; 4AP-HP, Hôpital Necker Enfants Malades, Service de Néphrologie–Transplantation Adulte, F-75015 Paris, France; 5AP-HP, Hôpital Européen Georges Pompidou, Service d’Urologie, F-75015 Paris, France; 6Réseau National pour Cancers Rares de l’Adulte PREDIR labellisé par l’INCa/AP-HP, Hôpital Bicêtre, F-94270 Le Kremlin-Bicêtre, France; 7AP-HP, Hôpital Européen Georges Pompidou, Service d’Anatomie et Cytologie Pathologiques, F-75015 Paris, France; 8Ecole Pratique des Hautes Etudes, EPHE, Université PSL, F-75014, France; 9UMR 9019-CNRS, Gustave Roussy Cancer Campus, F-94800 Villejuif, France; 10Service d’Urologie, Assistance Publique-Hôpitaux de Paris, Hôpital Bicêtre, F-94270 Le Kremlin-Bicêtre, France

**Keywords:** BHD syndrome, renal cell carcinoma, percutaneous thermal ablation, germline FLCN gene mutation, glomerular filtration rate

## Abstract

**Simple Summary:**

Percutaneous thermal ablation (TA) could be a safe and efficient nephron-sparing treatment for treating renal cell carcinoma (RCCs) associated with Birt–Hogg–Dubé (BHD) syndrome, a rare hereditary condition at greater risk of repeated treatments, even in the case of advanced chronic kidney disease, and should be systematically discussed as a treatment option. Indications for nephron-sparing management of these tumors in BHD patients depends on many factors, including size, number, and location of the tumor(s) in each kidney, patient surgical history, and renal function. The role of the tumor board including radiologists, interventional radiologists, urologists, and oncologists is essential.

**Abstract:**

BHD syndrome is characterized by an increased risk of bilateral and multifocal renal cell carcinoma (RCCs), but is rarely metastatic. Our report aims to analyze the outcome of patients with BHD syndrome who underwent percutaneous thermal ablation (TA). The present report included six BHD syndrome patients (five men) with a mean age of 66 ± 11 (SD) years who had a proven germline FLCN gene mutation and underwent TA for a renal tumor. Nineteen renal tumors (median two tumors per patient; range: 1–3), including seven chromophobe RCCs, five clear-cell RCCs, four papillary RCCs, two clear-cell papillary RCC, and one hybrid oncocytic/chromophobe tumor were treated in 14 ablation sessions. The mean size of the tumors was 21 ± 11 (SD) mm (median: 20 mm; interquartile range (IQR): 14–25 mm) for a mean volume of 7 ± 11 (SD) mL (median: 3; IQR: 1–5 mL). Technical success was achieved in all ablation sessions (primary success rate, 100%). The procedure was well tolerated under conscious sedation with no significant Clavien–Dindo complication (grade 2, 3, 4). All patients were alive with no distant metastasis during a median follow-up period of 74 months (range: 33–83 months). No local tumor progression was observed. The mean decrease in estimated glomerular filtration rate was 8 mL/min/1.73 m^2^. No patients required dialysis or renal transplantation. In this case series, percutaneous TA appeared as a safe and efficient nephron-sparing treatment for treating RCCs associated with BHD syndrome, even in the case of advanced chronic kidney disease.

## 1. Introduction

Birt–Hogg–Dubé (BHD) syndrome, initially described in 1977, is a rare autosomal dominant condition due to a germline mutation in the FLCN gene located on chromosome 17p11.2 identified in 2002 [1,2]. Today, several hundred families with BHD syndrome with FLCN mutations have been reported worldwide, but BHD syndrome is probably underdiagnosed because of the wide variability in its clinical expression [3]. BHD syndrome is characterized by skin fibrofolliculomas, multiple lung cysts, and spontaneous pneumothorax [2]. However, its most severe complication is a seven-fold increased risk of renal cell carcinoma (RCC), which tend to be bilateral and multifocal in more than half of patients, but is rarely metastatic [2,3,4,5]. Renal cancers occur in up to 30% of the patients during follow-up, at a mean age of 50 years (range 25–75 years) [3]. Renal tumors are more frequently indolent, such as chromophobe RCC (chRCC) and hybrid oncocytic/chromophobe renal tumors [3]. Other histological subtypes, such as clear-cell RCC (ccRCC) and papillary RCC (pRCC), including several mixed patterns have been also reported [3]. To date, no international guidelines have been established for the management of renal tumors in BHD syndrome patients [6]. It should take into account the indolent course of small renal tumors, the renal function preservation and the risk of multiple synchronous or metachronous tumors [7]. Percutaneous thermal ablation (PTA) is an effective treatment for RCC with minimal invasiveness and similar oncologic efficacy when the procedure can be repeated. Moreover, it allows repeated tumor ablation with minimal deterioration of the renal function [8,9]. It should, thus, be considered as an alternative option for patients with hereditary RCC who require repeated treatment for multiple tumors [10,11,12]. Previous reports have suggested the usefulness of radiofrequency (RFA) for RCC in patients with von Hippel–Lindau disease [10,11,12], but the efficacy of percutaneous thermal ablation (PTA) for RCC associated with BHD syndrome has been poorly studied [3,13]. The purpose of this report was to analyze the outcome of patients with BHD syndrome who underwent PTA (radiofrequency ablation (RFA), microwave ablation (MWA), and cryotherapy) of renal tumors.

## 2. Materials and Methods

Our study is a single-institution retrospective report based on our prospectively maintained database and was approved by the institutional review board (IRB number: CRM-2206-287).

### 2.1. Patients

All patients presenting with BHD syndrome and referred for PTA were included from January 2007 to May 2021 and are followed by the Reference Center of the National Network for Rare Cancers in Adult PREDIR (inherited predispositions to RCC) labeled by the French NCI (INCa). The PTA indication was approved by the institutional multidisciplinary tumor board. A renal biopsy was performed for six lesions of significant size above 1 cm. The remaining lesions exhibiting similar pattern at contrast-enhanced Computed Tomography (CE-CT) or contrast-enhanced Magnetic Resonance Imaging (CE-MRI) did not undergo renal biopsy and was considered as having similar pathology. If during follow-up one of the renal masses was growing faster, a targeted biopsy was performed. Patients lacking post-operative CT or MRI control were excluded as well. The patient was informed of the benefits and potential complications during a dedicated consultation and clinical findings were collected, including, the Eastern Cooperative Oncology Group (ECOG) status, the tumor history, the comorbidity factors, the renal status, the bleeding risk, and the results of the pre-procedural imaging examination. Written informed consent for PTA was obtained from all patients before initiating any procedure.

### 2.2. Procedure

The PTA technique (RFA, MWA, or cryoablation) was discussed during the interventional radiology meeting between three interventional radiologists with 5 to 20 years of experience in renal PTA. Pre-ablation blood tests included at least complete blood count, coagulation tests, and serum creatinine level.

### 2.3. Follow-Up

The follow-up imaging protocol consisted of both unenhanced and triphasic CE-CT and CE-MRI performed the following morning and at 2, 6, and 12 months after the procedure and then annually. Two senior radiologists with 5 to 25 years of experience in renal imaging reviewed each imaging examination for complication and potential local persisting disease. The appearance of a focal enhancing area within or adjacent to the ablation zone indicated local tumor progression [14,15]. Technical success was defined completing of the planned ablation protocol and complete coverage of the tumor by the ablation zone [14]. After each PTA procedure, the other renal masses were carefully followed.

Outcomes of thermal ablation were assessed, including primary success rate, complications, change in renal function, local tumor progression, development of metastases, survival rate after ablation, and global progression-free survival. Clinical success (primary success rate) was defined as no recurrence or metastasis. The median follow-up period was of 74 months (range: 33–83 months). Adverse events were graded according to the Clavien-Dingo classification [14]. Serum creatinine levels (sCr) (μmol/L) and estimated Glomerular Function Rate (eGFR) were recorded before each ablation and at last follow-up [16].

All information related to the procedure were prospectively collected, including the ablation technique, the number of needles/probes, and the duration of the procedure.

### 2.4. Descriptive Statistics

Categorical variables were reported as counts and percentages, and continuous variables as means (standard deviation) or median (range).

## 3. Results

### 3.1. Patients

Six patients (five men and one woman) with BHD syndrome and proven germline FLCN gene mutation were included. The mean age at referral for TA was 66 ± 11 (SD) years (median: 68 years; range: 52–84). Four patients (67%) had a history of partial nephrectomy (PN) for RCC on the kidney ipsilateral to the tumor, and one of them (17%) also had a total nephrectomy (TN) of the contralateral kidney. Two patients (33%) were treated with a curative dose of oral anticoagulants for ischemic heart disease with a coronary stent stopped 5 days before the PFA. The median ASA score was 3 (range: 2–4). Patient characteristics are summarized in Table 1.

### 3.2. Renal Tumors

A total of 19 renal tumors (median: 2 tumors per patient; range: 1–3) were treated using PTA on 14 ablation sessions. An 18-gauge needle biopsy was performed for 6 tumors (32%). For 13 masses, the CE-CT and/or CE-MRI was consistent with previous pathologically proven RCC for 13 tumors. Finally, seven chRCC, five ccRCCs, four pRCC, two clear-cell papillary renal tumors (ex clear-cell papillary RCC) (ccPRT), and 1 hybrid oncocytic/chromophobe tumor (HOCT) were included. The median size of the tumors was 20 ± 11 (SD) mm (mean: 21 mm; interquartile range (IQR): 14–25 mm) for a mean volume of 7 ± 11 (SD) mL (median: 3; IQR: 1–5 mL). The mean distance between the skin surface and the center of the lesion was 102 ± 15 (SD) mm (median: 98 mm; IQR: 91–115 mm). Fifteen renal tumors (79%) were located in the right kidney and four in the left, while six were more than 50% exophytic, one was less than 50% exophytic, and eleven entirely endophytic according to the RENAL Score [17]. Nine tumors (47%) were nearness to the collecting system or the sinus (< 4 mm), four (21%) were closed to the colon, three (16%) were in contact with the liver, and one (5%) was closed to the spleen and pancreas. All tumors were solid.

Renal tumor characteristics are summarized in Table 2.

### 3.3. Thermal Ablation Procedures

Seventeen RF ablations, 1 MW ablation, and 1 cryoablation were performed percutaneously under monitored anesthesia care (MAC) using US and CT guidance. Pain was controlled with local anesthesia and conscious sedation. Hydrodissection using 60 mL of 30% dextrose administered through a 20-Gauge needle inserted between the target tumor and the colon was used in 3 (16%) procedures to prevent bowel injury (Figure 1). Nine tumors were close to the collecting system or the sinus. For three of them, pyeloperfusion was performed using a 6 Fr ureteral stent to avoid thermal damage [18]. For the remaining lesions, a safety margin of more than 3 mm was found and considered to be sufficient according to our experience to avoid specific maneuvers.

RFA was performed using a single 17-gauge internally cooled electrode (Cool-tip™, Medtronic, UK) and a generator (E-series, Medtronic, UK). Depending on the tumor size, an electrode with an active tip length of 20 or 30 mm was used. After insertion of the electrode, a 6 to 12 min of ablation.

Cryoablation was conducted using an argon-based cryoablation system (Visual Ice™ Boston-Galil Medical Inc., St Paul, MN, USA) and 17-Gauge cryoprobes (IceRod™1.5 I-Thaw™, Boston-Galil Medical Inc., St Paul, MN, USA). The standard ablation protocol included two 10 min freezing cycles separated by 9 min of passive thawing and 1 min of active thawing. CT was performed to assess the size and position of the iceball at the end of each freezing cycle. The ablation procedure aimed to cover the target lesion with an ice ball margin larger than 5 mm, according to Georgiades et al. [19].

MWA was performed using a single 17-Gauge antenna (NeuWave™, Ethicon, Johnson & Johnson, New Brunswick, NJ, USA). The duration of the treatment was 8 min at 65 Watts.

Thermal ablation procedure characteristics are summarized in Table 2**.**

### 3.4. Efficiency and Adverse Effect

All tumors were successfully ablated after a single session with no imaging findings of persisting local tumors for all tumors at the end of the follow-up period (19/19, 100%). A single adverse event consisted of a subcapsular renal hematoma (52 × 48 × 21 mm) found incidentally at the post-procedural systematic CT. It did not result in a hemoglobin drop and did not require a blood transfusion (Clavien Dindo grade 0). No additional complications were reported.

Treatment outcome is presented in Table 3. Multiple synchronous and/or asynchronous tumors were treated in each patient (Figure 1). No local tumor progression was observed during a median follow-up period of 74 months (range: 33–83 months). All patients were alive with no distant metastasis during the same follow-up period.

Before thermal ablation, two patients had chronic kidney disease (CKD) stage 2 and three patients had CKD stage 3 (Figure 1 and Table 4). The median increase in sCr during follow-up was 13 μmol/L [2,3,4,5,6,7,8,9,10,11,12,13,14,15,16,17,18,19,20,21,22,23,24,25,26,27,28] corresponding to a median decrease in eGFR of 8 mL/min/1.73 m^2^ [4,5,6,7,8,9,10,11,12,13,14,15] (Figure 2 and Table 4). No patients developed end stage kidney disease.

## 4. Discussion

In this case series we reported six BHD syndrome patients who underwent TA for a renal tumor. Technical success was achieved in all ablation sessions and the procedure was well tolerated under conscious sedation with no significant Clavien–Dindo complication. All patients were alive with no distant metastasis during a median follow-up period of 74 months (range: 33–83 months), with no local tumor progression. The mean decrease in estimated glomerular filtration rate was 8 mL/min/1.73 m^2^ and no patients required dialysis or renal transplantation.

RCC occurs in up to 30% of patients with BHD syndrome [20]. Histological subtypes of BHD-associated renal tumors include mostly indolent tumors (HOCT, chromophobe RCC, oncocytoma), but also clear cell RCC and papillary RCC [21]. A close lifelong check-up is indicated for them and their relatives over 20 years old, as far as possible, to detect renal tumors [3,22,23]. The French National Cancer Institute Network PREDIR recommends MRI surveillance every 3 years with annual ultrasound surveillance in between [24]. To date, no international guidelines have been established so far for the clinical management of these patients [6].

Our report shows that PTA in patients with BHD syndrome is safe and provides good oncological control. The slow decrease in the renal function can be attributed to CKD, eventually accelerated by previous renal surgery or percutaneous ablation. The alteration of the renal function remains acceptable even in patients with multiple ablation procedures. It is important to note that all patients were alive with no distant metastasis at last follow-up and no dialysis was required.

Stamatakis L et al. [5] recommended abdominal imaging every 36 months in individuals without renal lesions at initial screening, and then, once a tumor is detected, follow-up imaging at regular intervals until the largest tumor reaches 3 cm in maximum diameter at which time nephron-sparing surgery should ideally be performed. Although the histology of renal tumors may vary in BHD, most tumors have a relatively indolent natural history and do not require adjuvant therapy if resected when localized in the kidney. This approach aims to achieve a curative oncologic outcome and to limit the impact on renal.

Percutaneous thermal ablation can be performed using several techniques, including radiofrequency ablation, microwave ablation, or cryoablation. The choice between these methods depends on the experience of the operator, the availability of the equipment as well as the size and position of the tumor and the cost of the probes. PTA can be conducted under conscious sedation, improving the tolerance of the procedure, particularly in the elderly population. As in any other hereditary renal tumors with multifocal, bilateral, asynchronous renal tumors such as von Hippel–Lindau disease, the key advantage of PTA is the reduced impact on renal function, even when compared to partial nephrectomy [25,26]. Moreover, the percutaneous TA can be repeated to treat other tumors without the main surgical limitation, i.e., the perirenal fibrosis that makes the dissection of the kidney extremely difficult [13].

Furthermore, Matsui et al. [13] and Gaillard et al. [26] showed that several lesions can be treated simultaneously. However, the need for multiple sessions versus a single intervention is debatable. Thanks to training and imaging performance, post-ablation imaging can detect persistence of viable tumor or local recurrence [27,28,29]. In addition, it appears that the only patients progressing to metastatic disease are those with ccCCR [5], which tends to be more aggressive than the more common oncocytic and chromophobe hybrid masses, which goes against aggressive management of the renal masses. Finally, it is notable that general anesthesia can be challenging in BHD patients with pulmonary cysts, particularly in the elderly BHD population. Indeed, excessive positive pressure ventilation is at risk of rupture of a pulmonary cyst which could lead to an associated tension pneumothorax [5]. Since PTA can be performed under conscious sedation, it has the advantage over surgery (performed under general anesthesia) in avoiding this risk.

The present report has some limitations. This monocentric retrospective report includes a limited number of tumors and patients, even though it is the largest cohort of BHD-associated renal tumors with Matsui et al. [13]. The ablation technique combined RFA, MWA, and cryoablation depending on the size and position of the tumors. However, most of the procedures were performed using RFA. Finally, the report was not designed to randomly compare partial nephrectomy and PTA efficacy and tolerance, and no information can be gathered for confirming the theoretical advantages of thermal ablation over surgery.

Indications for nephron-sparing management of these tumors in BHD patients depends on many factors, including size, number, and location of the tumor(s) in each kidney, patient surgical history, and renal function [3]. The role of the tumor board including radiologists, interventional radiologists, urologists, and oncologists is essential. Despite the lack of scientific data for discussing the appropriate timing of intervention [3,15,30], it is necessary to take into account that partial surgery can become very difficult after PTA in patients with new tumors or local tumor progression [5].

## 5. Conclusions

The present report confirms promising long-term oncologic and renal functional outcomes of PTA for BHD-associated multifocal renal tumors. Because of its minimal invasiveness and repeatability, PTA could be an effective and safe mini-invasive nephron-sparing treatment option for this rare hereditary condition at greater risk of repeated treatments and CKD.

## Figures and Tables

**Figure 1 cancers-14-04969-f001:**
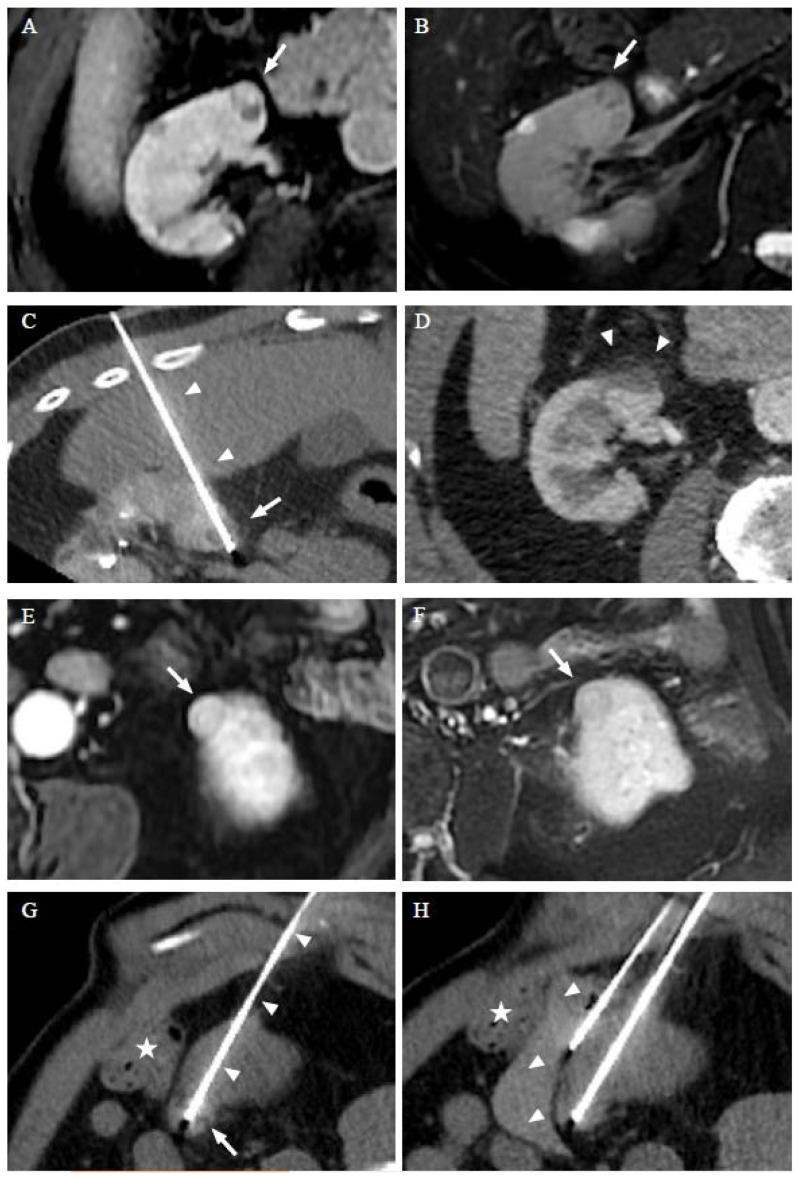
RFA of a 68-year-old man with Birt–Hogg–Dubé syndrome who had a history of partial nephrectomy and presents two ccRCCs treated in two separate sessions. Dynamic axial T1w sequences with fat suppression after gadolinium injection (**A**) and axial T2w image with fat saturation (**B**) show the first ccRCC (arrow) treated. Computed tomography (CT) images in the axial plane obtained during the RFA show the transhepatic probe (arrowheads) into the lesion (arrow) (**C**). CT images in the axial plane on the follow-up 6 months later show a RFA scar (arrow) (**D**). Dynamic axial T1w sequences with fat suppression after gadolinium injection (**E**) and axial T2w image with fant saturation (**F**) show the second ccRCC (arrow) developed in the left kidney 26 months after the initial RFA session. CT images in the axial plane obtained during the 2nd session of RFA show the RFA probe (arrowheads) into two of the lesions (**G**), and hydrodissection with 60 cc of serum glucose 30% was performed to protect the colon (star) (**H**). The patient was alive with no local progression or distant metastasis 33 months after the initial thermal ablation.

**Figure 2 cancers-14-04969-f002:**
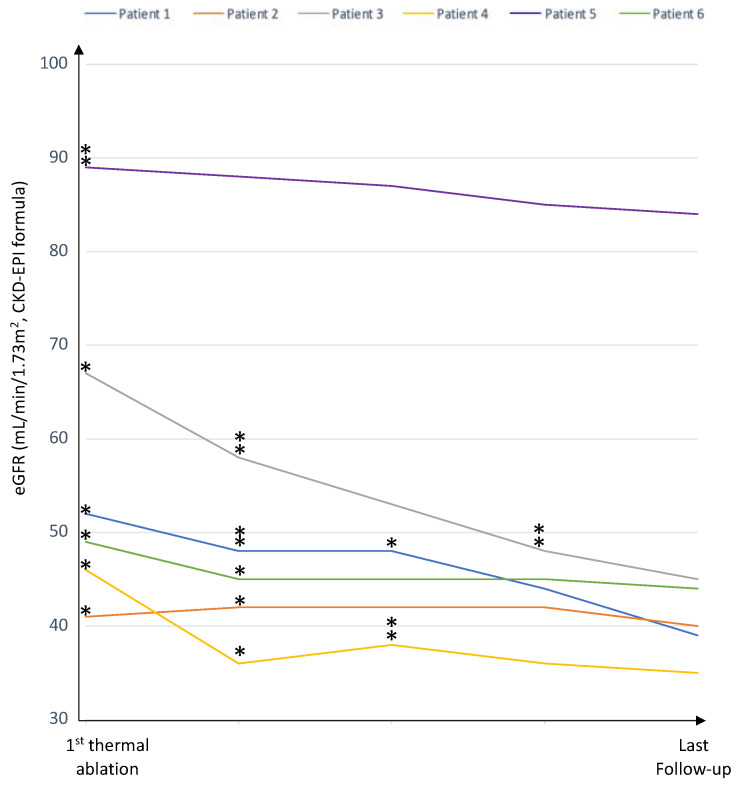
The renal function during the follow-up period for each of the six patients. * (one ablation) and ** (two ablations): Se Creatinine level at the moment of TA for each ablation procedure.

**Table 1 cancers-14-04969-t001:** Clinical, genetic, and histological characteristics of the six patients with renal cell carcinoma and Birt–Hogg–Dubé syndrome.

Patient #	Sex	FLCN Germline Mutation	Lung Cysts	History of Pneumothorax	History of Nephrectomy	Oral Anticoagulant	ASA	BMI	Age at Referral for Ablation (Year)	Tumor #	Histological Type of RCC
1	M	c.1285del, p.His429Thrfs*39	yes	yes	Partial	Yes	3	26	76	1	Chromophobe ^a^
										2	Chromophobe ^b^
										3	Chromophobe ^b^
									79	4	Chromophobe ^b^
2	M	c.1285del, p.His429Thrfs*39	yes	yes	No	Yes	4	26	84	5	Clear cell papillary renal tumor (ccPRT) ^a^
										6	Clear cell papillary renal tumor (ccPRT) ^b^
3	F	c.663dup, p.Met222Aspfs*26	yes	yes	Total* and partial	No	3	26	52	7	Chromophobe ^a^
										8	Clear cell RCC ^b^
										9	Clear cell RCC ^b^
										10	Clear cell RCC ^b^
										11	Hybrid oncocytic/chromophobe tumor (HOCT) ^a^
4	M	c.715C>T, p.Arg239Cys	yes	yes	Partial	No	2	29	64	12	pRCC ^b^
										13	pRCC ^b^
									70	14	pRCC ^b^
										15	pRCC ^b^
5	M	c.1300G>A, p.Glu434Lys	yes	no	Partial	No	2	30	59	16	Chromophobe ^a^
										17	Chromophobe ^b^
6	M	c.1579C>T, p.Arg527*	yes	no	No	No	2	24	68	17	ccRCC ^a^
									71	19	ccRCC ^b^

HOCT: Hybrid oncocytic/chromophobe tumor; cctpRCC: Clear cell papillary renal cell carcinoma. ^a^ Diagnosis by needle biopsy before ablation. ^b^ Diagnosis based on similar imaging pattern and previous pathology obtained from surgery or needle biopsy. * Single kidney unit.

**Table 2 cancers-14-04969-t002:** Characteristics of the renal cell carcinomas and thermal ablation procedures.

Tumor #	Renal Side	Localization	RENAL Score	Max. Diameter (mm)	Vol. (cc)	Nearness to the Collecting System or Sinus (<4 mm)	Nearness to the Digestive System	Nearness to Other Organ	Depth (mm)	Technique	Probe	Length (cm) /Exposure (mm)	Number of Probes	Number of Treated Tumors in the Same Session	Hydro- Dissection	Ureteral Stent
1	Right	<50% exophytic	7p	25	7.8	Yes	No	No	112	RFA	Cool-tip™ RFA Single 17 G	15/20	3	1	No	Yes
2	Right	entirely endophytic	10a	21	3.9	Yes	No	No	90	RFA	Cool-tip™ RFA Single 17-G	15/20	1	2	No	Yes
3	Right	entirely endophytic	10x	18	3.9	Yes	No	No	110	RFA	Cool-tip™ RFA Single 17-G	15/20	1	2	No	Yes
4	Right	< 50% exophytic	5x	22	2.9	No	Yes	No	107	RFA	Cool-tip™ RFA Single 17-G	15/30	2	1	Yes *	No
5	Right	≥ 50% exophytic	4x	37	24.1	No	No	No	70	RFA	Cool-tip™ RFA Single 17-G	15/30	3	1	No	No
6	Left	≥ 50% exophytic	5x	41	28.8	No	No	Spleen and pancreas	120	RFA	Cool-tip™ RFA Single 17-G	15/30	3	1	No	No
7	Left	< 50% exophytic	10a	54	37.2	Yes	No	Liver	95	Cryo	Galil IceSphere 1.5 17-G	17.5/30	8	1	No	No
8	Right	entirely endophytic	9p	13	1.1	Yes	Yes	Liver	120	RFA	Cool-tip™ RFA Single 17-G	15/20	1	2	No	No
9	Right	entirely endophytic	9a	13	1.0	Yes	No	No	90	RFA	Cool-tip™ RFA Single 17-G	15/20	1	2	No	No
10	Right	entirely endophytic	9x	15	1.0	Yes	No	No	95	RFA	Cool-tip™ RFA Single 17-G	15/20	1	2	No	No
11	Right	entirely endophytic	9a	14	0.7	Yes	No	No	92	RFA	Cool-tip™ RFA Single 17-G	15/20	1	2	No	No
12	Right	<50% exophytic	8p	21	4.4	Yes	No	No	92	RFA	Cool-tip™ RFA Single 17-G	15/20	1	1	No	No
13	Right	<50% exophytic	7a	24	6.0	No	Yes	No	85	MWA	NeuWaveTM PR Probe 17-G	20/NA	1	1	Yes *	No
14	Right	entirely endophytic	7p	26	4.8	No	No	No	98	RFA	Cool-tip™ RFA Single 17-G	15/30	1	2	No	No
15	Right	entirely endophytic	6p	15	1.8	No	No	No	86	RFA	Cool-tip™ RFA Single 17-G	15/20	1	2	No	No
16	Left	entirely endophytic	9x	20	2.5	No	No	No	126	RFA	Cool-tip™ RFA Single 17-G	20/20	1	2	No	No
17	Left	entirely endophytic	8a	11	0.4	No	Yes	No	118	RFA	Cool-tip™ RFA Single 17-G	15/20	1	2	No	No
18	Right	entirely endophytic	6a	10	0.4	No	No	Liver	101	RFA **	Cool-tip™ RFA Single 17-G	15/20	1	1	No	No
19	Right	< 50% exophytic	5a	12	0.8	No	Yes	No	125	RFA	Cool-tip™ RFA Single 17-G	20/20	1	1	Yes *	No

* hydrodissection with 60 mL of dextrose 30%. ** transhepatic ablation.

**Table 3 cancers-14-04969-t003:** Patient outcomes.

Patient #	Tumor #	Survival	Distant Metastasis	Follow-Up (Month)	Local Progression after Ablation	Complication
1	1	Alive	No	169	No	No
	2			167	No	No
	3			167	No	No
	4			137	No	No
2	5	Alive	No	49	No	No
	6			46	No	Subcapsular renal hemtoma
3	7	Alive	No	84	No	No
	8			81	No	No
	9			81	No	No
	10			74	No	No
	11			74	No	No
4	12	Alive	No	75	No	No
	13			32	No	No
	14			6	No	No
	15			6	No	No
5	16	Alive	No	22	No	No
	17			22	No	No
6	18	Alive	No	62	No	No
	19			33	No	No

**Table 4 cancers-14-04969-t004:** Evolution of the renal function of the six patients after each thermal ablation procedure.

Patient #	Thermo-Ablation	Serum Creatinine (μmol/L)	eGFR (mL/min/1.73 m)	ΔEGFR * (mL/min/1.73 m)
1	1st	117	52	13
	2nd and 3rd	125	48	
	4th	132	44	
	Last follow-up	137	39	
2	1st	135	41	1
	2nd	134	42	
	Last follow-up	135	40	
3	1st	85	67	22
	2nd and 3rd	96	58	
	4th and 5th	112	48	
	Last follow-up	116	45	
4	1st	139	46	11
	2nd	169	36	
	3rd and 4th	157	38	
	Last follow-up	166	35	
5	1st and 2nd	82	89	5
	Last follow-up	85	84	
6	1st	130	49	5
	2nd	137	45	
	Last follow-up	136	44	

Abbreviations: eGFR: estimated Glomerular Filtration Rate based on CKD-EPI equation. * Difference of eGFR before and after thermos-ablation procedure.

## Data Availability

The data presented in this study are available in this article.

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
