# Peer review of "Percutaneous Thermal Ablation for Renal Tumors in Patients with Birt–Hogg–Dubé Syndrome"

_cancers, 2022, doi:10.3390/cancers14204969_

Round 1

Reviewer 1 Report

cancers-1922509

Percutaneous thermal ablation for renal tumors in patients with  Birt-Hogg-Dubé syndrome

In this case series, 6 BHD syndrome patients with proven germline FLCN gene mutation who underwent TA for a renal tumor. 19 renal tumors including seven chromophobe RCCs, five clear-cell RCCs, four papillary RCCs, two clear-cell papillary RCC, and one hybrid oncocytic/chromophobe tumor were treated in 14 ablation sessions.

This manuscript only present 6 patients and meets only bare minimum of being a case series. So, I recommend the authors to remove the “study” term and stick to “report”.

Prior case series have also had similar or more sample size, strength, and limitation. Not sure about novelty of this case series.

Simple Summary:

Should focus on the main findings/ conclusion of the report, not background or aim.

Abstract:

Again, this is a case series, not study.

Conclusion is not supported by the sample size and analysis.

Introduction

OK.

The authors criticized that the efficacy of percutaneous thermal ablation for RCC associated with BHD syndrome has been poorly studied, but they have reported similar size of patients in their own case series.

Materials and Methods

-       This is not a study. This is only a case series.

-       Missing definition of technical success and clinical success.

Statistical analysis:

-       Kolmogorov-Smirnov test is used for large sample sizes, not a case series.

-       “chi square” not “Khi-2” or Fisher

-       Student t-test is meaningless for sample size of 6.

Results

Should reformatted and focus on case reports.

Discussion

Please start the discussion section with main findings.

Conclusions

The conclusion is not supported with the study design, sample size, and analysis/results.

Figures

Fine

Tables

Fine

Author Response

Sylvain Bodard

AP-HP, Hôpital Necker Enfants Malades

Service d’Imagerie Adulte

F-75015, Paris, France

+33618816210

[email protected]

To Pr Anthony Dohan

Collection Editor

Manuscript #: cancers-1922509Title: Percutaneous thermal ablation for renal tumors in patients with Birt-Hogg-Dubé syndrome

Dear Editorial Committee,

Thank you for considering our manuscript for publication in Cancers. We would like to thank the reviewers for their fruitful comments which help to improve the final manuscript.

Please find the revised manuscript considering all the queries/questions required by the reviewers.

Please find a point-to-point response to reviewer comments.

Thank you for your consideration of this manuscript.

Yours sincerely,

Dr. Sylvain Bodard

AP-HP, Hôpital Necker Enfants Malades

Service d’Imagerie Adulte

Université de Paris

Co-author

Pr Jean-Michel Correas

AP-HP, Hôpital Necker Enfants Malades

Service d’Imagerie Adulte

Université de Paris

Reviewer Comments:

Reviewer n°1 : 

In this case series, 6 BHD syndrome patients with proven germline FLCN gene mutation who underwent TA for a renal tumor. 19 renal tumors including seven chromophobe RCCs, five clear-cell RCCs, four papillary RCCs, two clear-cell papillary RCC, and one hybrid oncocytic/chromophobe tumor were treated in 14 ablation sessions.

This manuscript only present 6 patients and meets only bare minimum of being a case series. So, I recommend the authors to remove the “study” term and stick to “report”.

  • "study" was replaced by "report" in the entire manuscript.

Prior case series have also had similar or more sample size, strength, and limitation. Not sure about novelty of this case series.

  • This syndrome is extremely rare. In our reference center, we performedmore than 1500 renal tumor percutaneous ablations. In this rather large series, only 19 tumors were treated in patients with BHD syndrome. Although the series is limited, we believe that it contributes to the state of science on this subject. Moreover, the originality of our study is to have studied the impact on renal function, which is a major issue in this population. Moreover, all procedures were conducted under monitored anesthesia care (MAC) as procedural sedation analgesia.

Simple Summary:

Should focus on the main findings/ conclusion of the report, not background or aim.

  • We modified the Simple Summary: “Percutaneous TA could be as a safe and efficient nephron-sparing treatment for treating RCCs associated with BHD syndrome, even in the case of advanced chronic kidney disease, and should be systematically discussed as a treatment option.”

Abstract:

Again, this is a case series, not study.

  • We removed the term "study" and replaced it by the term "report".

Conclusion is not supported by the sample size and analysis.

  • We have qualified our conclusion by stating that the results were based on a case series only.

Introduction

OK.

The authors criticized that the efficacy of percutaneous thermal ablation for RCC associated with BHD syndrome has been poorly studied, but they have reported similar size of patients in their own case series.

  • This syndrome is extremely rare. In our center, we have experience with more than 1500 renal tumor removals, and among them, only 19 tumors in patients with BHD syndrome. Although the series is limited, we believe that it contributes to the state of science on this subject and the increase in the number of published cases should encourage the technique to be offered more systematically to all patients.

Materials and Methods

-       This is not a study. This is only a case series.

  • We have removed the term "study" and replaced it with the term "report".

-       Missing definition of technical success and clinical success.

  • Concerning the technical success, we have specified “ The appearance of a focal enhancing area within or adjacent to the ablation zone indicated local tumor progression [14–16]. Success was defined completing of the planned ablation protocol and complete coverage of the tumor by the ablation zone [15]”.
  • Concerning the clinical success, we have specified “ Clinical success (primary success rate) was defined as no recurrence or metastasis.”

Statistical analysis:

-       Kolmogorov-Smirnov test is used for large sample sizes, not a case series.

-       “chi square” not “Khi-2” or Fisher

-       Student t-test is meaningless for sample size of 6.

  • We removed all the statistical part not consistent with the small size of our series. We verified that our results were descriptive with no statistical analyses inappropriate in the case of case series.

Results

Should reformatted and focus on case reports.

  • We have checked that our results were only descriptive without statistical analyses not conforming to a series of cases.

Discussion

Please start the discussion section with main findings.

  • We started the discussion section with main findings: “In this case series, we reported six BHD syndrome patients who underwent TA for a renal tumor. Technical success was achieved in all ablation sessions and the procedure was well tolerated under MAC with no significant Clavien-Dindo complication. All patients were alive with no distant metastasis during a median follow-up period of 74 months (range: 33-83 months), with no local tumor progression. The mean decrease in estimated glomerular filtration rate was 8 mL/min/1.73 m2 and no patients required dialysis or renal transplantation.”

Conclusions

The conclusion is not supported with the study design, sample size, and analysis/results.

  • We have qualified our conclusion by stating that the results were based on a case series only: “ (…) PTA could be an effective and safe mini-invasive nephron-sparing treatment option (…)”

Figures

Fine

Tables

Fine

Reviewer 2 Report

This paper is very interesting and confirms the potential of thermal ablation.

Few suggestions.

Materials and methods

2.3 follow up (line 101) have you done both CE-CT and CE-MRI or CE-CT and/or CE-MRI?

Results:

 3.2 renal tumors: (line 151) six were exophytic and eleven were entirely endophytic. And the last one?

3.3 thermal ablation procedures: you described hydrodissection to prevent bowel injury, but you did't describe how you protected the excretory system in the nine tumors close to the collecting system or the sinus. Did you just insert a ureteral stent or did you perform pyeloperfusion (see for exemple Insights Imaging. 2017 Jun;8(3):357-363. doi: 10.1007/s13244-017-0555-4. Epub 2017 May 12.Tips and tricks for a safe and effective image-guided percutaneous renal tumour ablation. G Mauri 1 2, L Nicosia 3, G M Varano 4, G Bonomo 4, P Della Vigna 4, L Monfardini 5, F Orsi 4)?

Author Response

Sylvain Bodard

AP-HP, Hôpital Necker Enfants Malades

Service d’Imagerie Adulte

F-75015, Paris, France

+33618816210

[email protected]

To Pr Anthony Dohan

Collection Editor

Manuscript #: cancers-1922509Title: Percutaneous thermal ablation for renal tumors in patients with Birt-Hogg-Dubé syndrome

Dear Editorial Committee,

Thank you for considering our manuscript for publication in Cancers. We would like to thank the reviewers for their fruitful comments which help to improve the final manuscript.

Please find the revised manuscript considering all the queries/questions required by the reviewers.

Please find a point-to-point response to reviewer comments.

Thank you for your consideration of this manuscript.

Yours sincerely,

Dr. Sylvain Bodard

AP-HP, Hôpital Necker Enfants Malades

Service d’Imagerie Adulte

Université de Paris

Co-author

Pr Jean-Michel Correas

AP-HP, Hôpital Necker Enfants Malades

Service d’Imagerie Adulte

Université de Paris

Reviewer Comments:

Reviewer n°2 :

This paper is very interesting and confirms the potential of thermal ablation.

Few suggestions.

Materials and methods

2.3 follow up (line 101) have you done both CE-CT and CE-MRI or CE-CT and/or CE-MRI?

  • Both CE-CT and CE-MRI were performed. We specified this point : “ (…) The follow-up imaging protocol consisted of both unenhanced and triphasic CE-CT and CE-MRI performed the following morning and (…)”

Results:

 3.2 renal tumors: (line 151) six were exophytic and eleven were entirely endophytic. And the last one?

  • The last one was partially exophytic (<50) according to the RENAL Score. We specified this point: “(…) six were more than 50% exophytic, one was less than 50% exophytic and 11 entirely endophytic according to the RENAL Score (…)”

3.3 thermal ablation procedures: you described hydrodissection to prevent bowel injury, but you did't describe how you protected the excretory system in the nine tumors close to the collecting system or the sinus. Did you just insert a ureteral stent or did you perform pyeloperfusion (see for exemple Insights Imaging. 2017 Jun;8(3):357-363. doi: 10.1007/s13244-017-0555-4. Epub 2017 May 12.Tips and tricks for a safe and effective image-guided percutaneous renal tumour ablation. G Mauri 1 2, L Nicosia 3, G M Varano 4, G Bonomo 4, P Della Vigna 4, L Monfardini 5, F Orsi 4)?

  • In order to protect the collecting system from thermal damage, a 6 Fr ureteral stent was inserted for three of the nine tumors close to the renal sinus, before ablation. Pyeloperfusion was performed with chilled saline infusion during treatment according to recommendations . For the remaining lesions, a safety margin of more than 3 mm was considered to be sufficient according to our experience to avoid specific maneuvers. We specified this point in the manuscript: “Nine tumors were close to the collecting system or the sinus. For three of them, pyeloperfusion was performed using a 6 Fr ureteral stent to avoid thermal damage. For the remaining lesions, a safety margin of more than 3 mm was found and considered to be sufficient according to our experience to avoid specific maneuvers.”

Round 2

Reviewer 1 Report

The text should be checked out for grammatical errors.